# Helminth Seropositivity Inversely Correlated with Th1 and Th17 Cytokines and Severe COVID-19

**DOI:** 10.3390/vaccines13030252

**Published:** 2025-02-27

**Authors:** Brice Armel Nembot Fogang, Julia Meyer, Linda B. Debrah, Michael Owusu, George Agyei, Derrick Adu Mensah, John Boateng, Jubin Osei Mensah, Ute Klarmann-Schulz, Sacha Horn, Inge Kroidl, Ezekiel Bonwin Ackah, Richard O. Phillips, Augustina Sylverken, Alexander Y. Debrah, Achim Hoerauf, Tomabu Adjobimey

**Affiliations:** 1Department of Clinical Microbiology, School of Medical Sciences, Kwame Nkrumah University of Science and Technology, Kumasi 00233, Ghana; 2Institute of Medical Microbiology, Immunology and Parasitology (IMMIP), University Hospital Bonn, 53127 Bonn, Germany; 3Kumasi Centre for Collaborative Research in Tropical Medicine (KCCR), Kwame Nkrumah University of Science and Technology, Kumasi 00233, Ghana; 4German West African Center for Global Health and Pandemic Prevention (G-WAC), Kumasi 00233, Ghana; 5Department of Medical Diagnostics, Kwame Nkrumah University of Science and Technology, Kumasi 00233, Ghana; 6Department of Medical Laboratory Technology, Royal Ann College of Health, Kumasi P.O. Box KS 6253, Ghana; 7Department of Public Health Education, Akenten Appiah-Menka University of Skills Training and Entrepreneurial Development, Kumasi P.O Box 1277, Ghana; 8Department of Pathobioogy, School of Veterinary Medicine, Kwame Nkrumah University of Science and Technology, Kumasi 00233, Ghana; 9Bonn-Cologne Site, German Center for Infectious Disease Research (DZIF), Bonn, Germany; 10Institute of Infectious Diseases and Tropical Medicine, University Hospital, Ludwig-Maximilians-Universität Munich, 80539 Munich, Germany; 11Munich Site, German Center for Infectious Disease Research (DZIF), Munich, Germany; 12Faculty of Allied Health Sciences, Kwame Nkrumah University of Science and Technology, Kumasi 00233, Ghana; 13Faculté des Sciences et Techniques (FAST), Université d’Abomey Calavi, Abomey Calavi 05 BP 1604, Benin

**Keywords:** SARS-CoV-2, helminth, co-exposure, Africa, clinical outcomes, asymptomatic

## Abstract

**Background/Objectives:** The COVID-19 pandemic has significantly impacted global health. However, Africa has reported relatively low numbers of cases and fatalities. Although the pandemic has largely receded, the reasons for its milder course on the African continent have not yet been fully clarified. This study explored the hypothesis that helminth co-infections may have contributed to these observations. **Methods:** A retrospective cohort study was conducted using 104 plasma samples collected during the third wave of the pandemic in the Ashanti Region of Ghana. Luminex assays were used to measure SARS-CoV-2-specific IgA and IgG, neutralizing antibodies, systemic cytokines and helminth-specific IgG. **Results:** The results indicated that the highest cumulative seroprevalence of helminths (61.5%) was observed in asymptomatic COVID-19 patients. In comparison, mild and moderate patients had helminth seropositivity rates of 43.8% and 34.5%, respectively, which were 1.4 and 1.8 times lower than those of the asymptomatic group, respectively. Notably, the two severe COVID-19 cases investigated were seronegative for all three of the helminths tested. Strikingly, co-exposure resulted in lower SARS-CoV-2-specific IgA/IgG expression and reduced neutralization potential. However, co-seropositive individuals for helminths and SARS-CoV-2 exhibited a higher expression of Th2 cytokines and IL-10 over Th1 cytokines compared to SARS-CoV-2-positive individuals alone. **Conclusion:** These data suggest that co-exposure to helminths could mitigate the severity of COVID-19 outcomes by reducing the Th1 and Th17 responses; this highlights the potential protective role of helminthiasis against severe COVID-19. These findings provide valuable insights for the development of public health policies in helminth-endemic regions and underscore the importance of considering helminth co-infections in managing viral infections. It also offers a plausible explanation for the milder disease severity observed in helminth-endemic regions while raising critical considerations regarding vaccine efficacy, as helminth-induced immune modulation may influence the magnitude and quality of vaccine-induced immune responses.

## 1. Introduction

Since its emergence in Wuhan, China, in late 2019, the COVID-19 pandemic, caused by SARS-CoV-2, has significantly impacted global health and economies [1]. As the virus started to spread globally, healthcare professionals faced a plethora of clinical presentations and outcomes, ranging from asymptomatic infections to severe acute respiratory distress and death [2]. Governments implemented various public health measures, including lockdowns, social distancing, and vaccination campaigns to mitigate the spread of the virus. Despite these efforts, the pandemic has resulted in 775 million confirmed cases and more than 7 million deaths as of 7 July 2024 [3]. Older age and comorbidities, such as hypertension, diabetes, cardiovascular diseases, chronic respiratory conditions, and obesity, were found to be associated with severe clinical outcomes [4].

A striking aspect of the pandemic is the variation in cases and fatality rates across regions. In contrast with high-income countries, such as the United States and several European nations, the trajectory of the pandemic in Africa has been unexpectedly mild, with lower case numbers and fatality rates [5,6]. While it is widely accepted that younger demographics on the continent have contributed to the milder impact of the pandemic, emerging hypotheses suggest that the high prevalence of helminth infections in many African populations may also have played a role [7,8,9].

Helminths are parasitic worms that are widespread in many regions of the world, particularly in sub-Saharan Africa [10]. At least 1.5 billion people are thought to be infected with helminths, which include lymphatic filariae, schistosomes, and soil-transmitted helminths, primarily in sub-Saharan Africa, leading to significant health issues such as malnutrition, anemia, and chronic illness, as well as perpetuating cycles of poverty in these regions [10]. In addition, helminths are renowned for their ability to dampen pro-inflammatory responses and promote anti-inflammatory pathways, skewing their host’s immune system towards Th2 and regulatory responses, including cytokines such as IL-4, IL-5, IL-13, and TGF-β [11,12,13,14]. Through these unique properties, they can potentially affect the course of viral co-infections, such as SARS-CoV-2, altering the severity and progression of diseases.

The impact of helminth co-infection on the clinical outcomes of viral infections has not yet been fully elucidated [15]. Emerging hypotheses suggest that helminth co-infections can attenuate the severity of COVID-19 through their immunomodulatory properties [16]. Previous studies have demonstrated that helminths can significantly affect host susceptibility and the clinical outcomes of viral and bacterial infections [17,18,19,20,21]. A recent in vitro study showed that the activation of SARS-CoV-2 reactive CD4+ T helper cells was significantly reduced by helminth antigens. The study examined the impact of antigens extracted from three major helminth parasites, *Onchocercid volvulus*, *Ascaris lumbricoides* and *Brugia malayi*, on immune reactivity to SARS-CoV-2 peptides [22,23]. However, the flip side has potential disadvantages, as helminth infections have been associated with diminished vaccine efficacy, increased susceptibility to concurrent infections, and potential interference with tumor immunosurveillance [18,24,25,26,27,28,29,30,31,32,33,34].

One potential explanation is that helminth infections may trigger a Th2-skewed and regulatory immune response, which in turn inhibits or regulates the Th1 immune response. Suppression of the Th1 immune response can also dampen the hyperinflammation linked to severe COVID-19, resulting in improved health outcomes [9,15,26,35,36].

Building on these findings, the present study aimed to elucidate the immunomodulatory effects of helminths on SARS-CoV-2-specific Th1 and Th17 immune responses and their associated clinical outcomes in an African population. This study also examined the proportion of asymptomatic COVID-19 cases among individuals concurrently exposed to helminth antigens. By examining the relationship between helminth seropositivity, cytokine profiles, and disease severity, we sought to shed light on the complex interplays between helminth co-infection and COVID-19 severity. This investigation promises to enhance our understanding of the dynamic interplays between parasitic infections and viral immunity, offering new insights into the immune regulation determinants of disease outcomes and shaping public health policies in helminth-endemic areas.

## 2. Materials and Methods

### 2.1. Participants, Sample Collection, and Ethics

A retrospective cohort cross-sectional study was conducted using 104 archived plasma samples from the Ashanti Region collected during the third wave of the pandemic from June 2022 to December 2022. During this period, the Delta variant (B.1.617.2) was the predominant strain in circulation in Ghana [37]. To avoid bias and ensure representativeness, all participants, including a diverse patient population across the region and reflecting the region’s demographics, were included in the study. Although the initial recruitment goal was to recruit 200 participants, the final sample size was 104 participants because of the low incidence of active COVID-19 infections in the region. Recruitment was closed at the end of the wave in Ghana to maintain consistency.

The recruitment was conducted by teams from the Kumasi Center for Collaborative Research in Tropical Medicine (KCCR) in Ghana. All personnel involved in sample collection were thoroughly trained and provided with standardized protocols. The demographic and clinical data of all the participants were collected using predefined questionnaires. Blood samples were collected in EDTA tubes under aseptic conditions and subsequently processed for plasma by density gradient centrifugation as previously described [38]. The plasma was then aliquoted into tubes and stored at −20 °C before being shipped to the Bonn University Hospital, Institute of Medical Microbiology, Immunology, and Parasitology (IMMIP) for further analysis. Samples were collected and prepared according to the guidelines of the Committee on Human Research and Publication and Ethics at the School of Medical Sciences, Kwame Nkrumah University of Science and Technology, Kumasi, Ghana (reference CHRPE/AP/072/21). Ethical approval for the investigations performed in Germany was obtained from the Ethics Committee of the University Hospital Bonn (Lfd. Nr. 439/20).

Participants with underlying diseases or conditions, such as high blood pressure, malaria, HIV, type 2 diabetes, and pregnancy, which could potentially confound the study’s outcome measures were excluded. These conditions may influence immune response and other physiological parameters, thereby affecting the interpretation of the results. By adhering to these criteria, the study aimed to maintain a homogeneous participant pool, ensuring that the observed effects were primarily attributable to the presence of helminths and/or a SARS-CoV-2 infection.

### 2.2. Quantification of SARS-CoV-2-Specific IgA and IgG and Severity Classification

Enzyme-linked immunosorbent assays (ELISAs) were performed at the IMMIP, University Hospital Bonn, Germany. The quantification of SARS-CoV-2-specific IgA and IgG was performed using the Euroimmun SARS-CoV-2 IgG/IgA ELISA kit (Euroimmun, Lübeck, Germany). The procedure was automated using the Euroimmun analyzer I, adhering to the manufacturer’s instructions and following the protocol previously described [39]. Optical density (OD) was measured at 450 nm. The expressions of SARS-CoV-2-specific IgG and IgA were calculated, and the results were interpreted according to the manufacturer’s protocol. Initial testing by PCR, as previously described [40], was used to grade COVID-19 severity according to the National Institutes of Health (NIH) guidelines as follows: Asymptomatic infection refers to individuals who tested positive for SARS-CoV-2 by PCR yet exhibited no symptoms indicative of COVID-19. Mild illness denotes those displaying any signs and symptoms of COVID-19 (e.g., cough, loss of taste, malaise, fever, headache, myalgia, nausea, vomiting, sore throat, diarrhea, and smell) without experiencing shortness of breath or dyspnea. Moderate illness pertains to patients exhibiting signs of lower respiratory disease during clinical evaluation, with oxygen saturation levels (SpO2) of ≥94% on room air at sea levels as assessed by pulse oximetry. Severe sickness pertains to patients exhibiting SpO2 levels below 94% on room air at sea level, a PaO2/FiO2 ratio under 300 mm Hg, a respiratory rate above 30 breaths per minute, or lung infiltration of more than 50% [41].

### 2.3. Quantification of Helminth-Specific IgG

*Ascaris lumbricoides*, *Strongyloides ratti*, and *Acanthocheilonema viteae* IgG ELISA Kits (DRG Instruments GmbH, Marburg, Germany and Bordier Affinity Products, Chantanerie, Switzerland) were used to detect specific IgG in the plasma of recruited individuals. All samples were examined for *Strongyloides ratti*, *Acanthocheilonema viteae*, and *Ascaris lumbricoides*. *Strongyloides ratti* was used as a homologous antigen for human *Strongyloides* species, and *Acanthocheilonema viteae* was used as a homologous antigen for human filarial nematode species. The procedure was performed manually in accordance with the manufacturer’s protocol as previously documented [17]. Optical density was assessed at wavelengths of 450 and 620 nm.

### 2.4. Quantification of Neutralizing Antibody Levels

To assess the levels of neutralizing antibodies specific to SARS-CoV-2 variants, we employed the SARS-CoV-2 Variants Neutralizing Antibody 6-Plex ProcartaPlex Panel (Thermo Fisher Scientific, Waltham, MA, USA), following the manufacturer’s protocol as previously outlined [17]. This assay enables the detection and quantification of SARS-CoV-2 neutralizing antibodies in serum and plasma, allowing for a direct comparison of neutralizing antibody levels against the following five SARS-CoV-2 variants: the wild-type and variants B.1.351 (β), B.1.617.2 (δ), P1 (γ), B.1.1.529 (ο), and B.1.1.7 (α). The samples were analyzed using a MagPix Luminex instrument. Data analysis was conducted using the ProcartaPlex Analyst application (Thermo Fisher Scientific) and the provided neutralization (%) equation.

### 2.5. Quantification of Systemic Cytokine and Chemokine Expression

The Cytokine Storm 21-plex Human ProcartaPlex Panel (Thermo Fisher Scientific) was utilized to evaluate systemic cytokine expression, following the manufacturer’s guidelines, as previously detailed [17]. This panel comprises 21 cytokines and markers linked to cytokine release syndrome (CRS), including IFN alpha, GM-CSF, IFN gamma, IL-1 beta, IL-4, IL-5, IL-10, MIP-1 alpha (CCL3), IL-6, IL-8 (CXCL8), IL-12p70, IL-13,IL-2, IL-17A (CTLA-8), IL-18, IP-10 (CXCL10), MCP-1 (CCL2),G-CSF (CSF-3), MIP-1 beta (CCL4), TNF beta, and TNF alpha. Cytokine quantification was conducted via the MagPix Luminex apparatus. The ProcartaPlex ™ online analysis program (Thermo Fisher Scientific) was utilized for data analysis.

### 2.6. Statistical Analysis

The primary objective of this study was to examine the proportion of asymptomatic COVID-19 cases among individuals concurrently exposed to helminth antigens. The baseline characteristics were reported as the median with interquartile range for continuous variables such as age, *Ascaris lumbricoides*-, *Strongyloides ratti*-, and *Acanthocheilonema viteae*-specific IgG, and as proportions for categorical variables such as sex and infection status (mono- and co-exposure). The χ^2^ test was used to compare the proportions across different categories of COVID-19 severity, providing additional insights into the distribution of categorical variables. The differences in helminth-specific IgG and cytokine expression among the severity groups were analyzed using the Kruskal–Wallis test. Due to the small number of severe cases in our study, the moderate and severe cases were combined for the analysis. The Mann–Whitney U test was used to compare cytokine expression and SARS-CoV-2-specific IgA/IgG between the mono- and co-exposed groups. The data analysis was performed using SPSS V. 26. All graphs were generated using GraphPad Prism version 10.2.0. (La Jolla, CA, USA).

## 3. Results

### 3.1. Demographic and Clinical Characteristics

The demographic and clinical characteristics of the study population are summarized in Table 1. The study included 52.9% (95% CI: 42.8–62.8%) males and 47.1% (95% CI: 37.2–57.2%) females, aged 13–90 years. The median age of the participants was 33 years (IQR: 20–48). The most participative age group consisted of individuals aged between 11 and 22 years, accounting for 28.8% (95% CI: 20.4–38.6%), while the least were individuals aged between 44 and 54 years [10.6% (95% CI: 5.4–18.1%)]. Of the 104 participants, 29.8% (95% CI: 21.2–39.6%) were healthy, 25.0% (95% CI: 17.0–34.4%) were asymptomatic, 15.4% (95% CI: 9.1–23.8%) were mild, 27.9% (95% CI: 19.5–37.5%) were moderate, while only 1.9% (95% CI: 0.2–6.8%) exhibited severe SARS-CoV-2-related symptoms. Moderate/severe cases were predominantly observed in the older age groups, suggesting a potentially higher risk of severe infection among older individuals (≥55) compared to younger age groups (11–22).

### 3.2. Assessment of SARS-CoV-2 and Helminth Co-Exposure Rates

A SARS-CoV-2 seroprevalence rate of 52.9% (95% CI: 42.8–62.8%) (Figure 1A) and helminth seroprevalence rate of 46.2% (95% CI: 36.3–56.2%) (Figure 1B) were observed. *Ascaris lumbricoides* emerged as the predominant helminth exposure at 40.4% (95% CI: 30.9–50.5%), whereas the least frequent helminth seropositivity was against *Strongyloides ratti* [2.9%, (95% CI: 0.6–8.2%)]. Cases of multiple helminth seroreactivity were observed only between *A. limbricoides* and *S. ratti* at 0.96% (95% CI: 0.0–5.2%) (Figure 1B). A total of 50.0% (95% CI: 35.2–64.8%) of helminth-seropositive individuals were also SARS-CoV-2-seropositive, whereas 43.6% (95% CI: 30.3–57.7%) of SARS-CoV-2-seropositive individuals were also helminth seropositive (Figure 1C). The seroprevalence of SARS-CoV-2-specific IgA [20.19% (95% CI: 13.6–28.9%)] was significantly higher than SARS-CoV-2-specific IgG [1.9% (95% CI: 0.05–6.74%)] (Figure 1A). This pattern may indicate recent or ongoing exposure to SARS-CoV-2, particularly at the mucosal surfaces where IgA is rapidly produced. Low IgG suggests that systemic immunity has not yet fully developed. This is often observed during the early phase of the infection when the immune system prioritizes the local containment of the pathogen.

### 3.3. Helminth-Exposed Individuals Exhibit Asymptomatic COVID-19 Clinical Outcomes

To determine the influence of helminth exposure on COVID-19 severity, a multivariate regression model was used. Age was found to have a significant impact on disease severity, with older individuals having higher odds of severe COVID-19 outcomes (OR = 1.046, *p* = 0.003). After adjusting for age and sex, exposure to helminths was significantly associated with a strong protective effect against severe COVID-19 outcomes, with non-exposed individuals having a higher risk of developing more severe COVID-19 outcomes than exposed individuals (OR = 4.115, *p* = 0.018) (Table 2). Specifically, *A. lumbricoides* exposed individuals had a lower risk of developing severe COVID-19 than non-exposed individuals (OR = 4.038, *p* = 0.028). The other helminths were not statistically significant.

### 3.4. Helminth Seroprevalence Inversely Correlates with COVID-19 Severity

To further analyze the relationship between helminth seropositivity and COVID-19 severity, we investigated the prevalence of helminth seropositivity across different levels of COVID-19 severity. The seroprevalence of *A. lumbricoides* was significantly higher than those of *S. ratti* and *A. viteae* (homologous of multiple human filarial species). The results further indicated that the highest cumulative seroprevalence of helminths (61.5%) was observed in the asymptomatic infected SARS-CoV-2 group. In contrast, mild and moderate COVID-19 patients exhibited 1.4 (43.8%) and 1.8 (34.5%) times lower helminth seropositivity than the asymptomatic group. Notably, the two severe COVID-19 cases investigated in this study were seronegative for all the three helminths tested (Figure 2).

### 3.5. Elevated Helminth-Specific IgG Levels Correlate with Asymptomatic SARS-CoV-2 Infection

Next, we compared the relative expression levels of helminth-specific IgG in asymptomatic, mild, and moderate/severe cases. The data show that helminth-specific IgG indices were significantly higher in asymptomatic SARS-CoV-2-infected individuals (Asy) than in moderate and severe COVID-19 patients (Mod/Sev). The results for all the helminths tested followed this trend, peaking in asymptomatic individuals and diminishing with increasing disease severity. Asymptomatic individuals exhibited 1.5 times the levels of helminth-specific IgG compared to those with moderate/severe symptoms (Figure 3).

### 3.6. Low Expression of SARS-CoV-2-Specific IgA/IgG and Neutralization Potency in Helminth-Co-Exposed Individuals

Furthermore, we aimed to identify the influence of helminth exposure on the development of humoral immunity against SARS-CoV-2. The results indicated that individuals exposed to helminths showed a significantly reduced expression of SARS-CoV-2-specific IgA/IgG (*p* = 0.0099, *p* = 0.0260) (Figure 4A,B) and neutralization potency against the wild type and variants (Figure 5). Similarly, helminth-specific IgG was negatively correlated with SARS-COV-2 neutralizing antibody expression (Figure 4C).

### 3.7. Lower Expression of Th1 and Th17 and Elevated Expression of Th2 Cytokines in Helminth Co-Exposed Individuals

To determine the immunological basis underlying the association between helminths and SARS-CoV-2 co-exposure, we used the Mann–Whitney U test to compare systemic Th1, Th2, and Th17 cytokines between SARS-CoV-2 mono-exposure (n = 31) and helminth co-exposure (n = 24). The results are shown in Figure 5. Co-exposed individuals exhibited elevated IL-10, IL-13, and IL-5 expression compared with mono-exposed individuals (*p* < 0.0001, *p* < 0.0001, and *p* = 0.0002, respectively). In contrast, mono-exposed participants displayed significantly higher expression of Type 1 and Th17 pro-inflammatory cytokines, including G-CSF, GM-CSF, IFN alpha, IFN gamma, IL-1 β, IL-2, IL-6, IL-8, IL-17A, IL-18, and IP-10 (*p* = 0.0132, 0.0464, <0.0001, 0.0009, 0.0263, 0.0522, 0.0299, 0.0231, 0.0040, and 0.0278, respectively) (Figure 6).

### 3.8. Reduced Expression of Th1 and Th17 Cytokines in Asymptomatic SARS-CoV-2 Exposed Individuals

To better understand the immunological differences across the spectrum of COVID-19 clinical presentations, we analyzed the systemic expression of different cytokines in the COVID-19 clinical groups. The data revealed a distinct cytokine profile associated with disease severity. Asymptomatic individuals exhibited a significantly higher expression of Th2 cytokines compared to patients with moderate and severe COVID-19, specifically IL-5 (*p* = 0.0444) and IL-13 (*p* = 0.0389). Conversely, they displayed significantly lower levels of Th1- and Th17-related pro-inflammatory cytokines, including IFN-γ (*p* = 0.0158), IFN-α (*p* = 0.0004), IL-6 (*p* = 0.0008), IL-8 (*p* = 0.0046), IL-17 (*p* = 0.0036), IL-12 (*p* = 0.0393), IL-18 (*p* = 0.0006), and IP-10 (*p* = 0.0004). In contrast, moderate to severe COVID-19 cases showed reduced expression of Th2 cytokines and notably higher levels of Th1 and Th17 cytokines. Specifically, these patients exhibited significantly higher levels of IFN-γ, IFN-α, IL-6, IL-8, IL-17, IL-12, IL-18, and IP-10 compared to asymptomatic and mild cases. However, the levels of other cytokines, including G-CSF (CSF-3), GM-CSF, IL-1β, IL-4, IL-2, MCP-1 (CCL2), MIP-1α (CCL3), TNF-α, MIP-1β (CCL4), and TNF-β, were not significantly different across the groups. These findings highlight the critical role of cytokine regulation in determining the clinical outcomes of COVID-19 (Figure 7).

## 4. Discussion

The examination of the demographic data in our study corroborates the preliminary findings indicating that advanced age and male gender contribute to severity of COVID-19 [17,42,43,44,45,46,47]. After adjusting for age and sex, our main findings support the hypothesis that co-exposure to helminths significantly leads to asymptomatic SARS-CoV-2 clinical outcomes. They also demonstrate that the presence of helminth antigens modulates the immune responses to SARS-CoV-2 in such a way that pro-inflammatory cytokine responses, antibody responses (IgA and IgG), and SARS-CoV-2 CD4+ reactive T cells are reduced, leading to better health outcomes despite the reduction in humoral immunity. This suggests that previous and current helminth exposure may reduce the risk of developing severe COVID-19 [8,17,48].

Helminths induce gut microbiome changes, which can affect the host’s immune response. Therefore, they can likely influence SARS-CoV-2 pathogenesis in the following two different ways: either by directly modulating the immune system or by influencing the microbiome balance [49,50,51]. Interactions between enteric helminths and microbiota have been shown to protect against pulmonary viral infections in animal models [50], and this protection has been proven effective. Understanding the effects of environmental pathogen exposure and lifestyle on the immune system can provide a better understanding of the effects of helminth exposure on COVID-19 clinical outcomes as well as the differential course of the pandemic worldwide [20]. These factors have been proven to affect the immune system and inflammatory response, particularly in low- and medium-income countries where helminths are endemic [52,53,54,55,56]. These parasites can affect immune system functionality, modulating the immune response to SARS-CoV-2, causing COVID-19 to spread differently in different regions of the world. While it is important to acknowledge the potential influence of other pathogens, environmental factors, and demographic factors, such as age, sex, and BMI, on COVID-19 clinical outcomes, our study suggests that helminths are a key contributing factor. This assertion is corroborated by global epidemiological data, as regions with high helminth endemicity have frequently demonstrated lower rates of severe COVID-19 cases and mortality compared to areas with low helminth endemicity [35]. Moreover, experimental studies have demonstrated that helminth infections can modulate the host immune response in a way that may mitigate the severity of other infectious diseases, including respiratory infections [12,13,14,38,57]. Simultaneously, we found that helminth-exposed individuals exhibited a lower expression of SARS-CoV-2-specific immunoglobulins (IgA and IgG) and a lower neutralizing potential. This result indicates that co-exposure to helminths leads to a diminished humoral response to SARS-CoV-2 and a reduced efficacy of antibodies in binding to the virus’ receptor binding domain (RBD). This result suggests that co-exposure to helminths leads to a diminished humoral response to SARS-CoV-2, as well as the reduced efficacy of antibodies in inhibiting the receptor-binding domain (RBD) of the virus. At first glance, this may seem contradictory, as antibody-mediated immune responses are typically recognized for their crucial role in viral clearance and in enhancing overall health outcomes. However, in the context of SARS-CoV-2 infection, disease severity is more closely linked to an excessive inflammatory response, also known as a cytokine storm, rather than a weak humoral response [57,58,59,60,61]. If co-exposure to helminth antigens modulates the cytokine storm, it is plausible that the overall health outcome might be better despite low immunoglobulin levels [60]. Additionally, moderate levels of neutralizing antibodies may still be adequate to clear the infection. On the other hand, it also indicates that helminth antigen exposure may diminish vaccine efficacy against certain pathogens [15,26,27,28,31,62,63]. Our findings, while preliminary, illustrate the vital role of considering exposure to multiple pathogens during clinical trials and how essential it is to balance the advantages and disadvantages introduced by helminth co-endemicity when considering health outcomes. This is particularly important in populations already burdened by neglected tropical diseases (NTDs), such as those in low-income and middle-income countries.

Co-exposed individuals showed lower expression levels of Th1 and Th17 cytokines, including G-CSF, IFN-α, IFN-γ, IL-1 β, IL-17A, IL-18, IP-10, IL-6, IL-2, and GM-CSF, while upregulating the expression of Th2 cytokines such as IL-5, IL-10, and IL-13. The higher expression of Th2 cytokines and IL-10 might be the reason for the low expression of Th1 and Th17 cytokines, which may lead to asymptomatic outcomes. IL-10 is widely recognized as a powerful immune-regulatory cytokine [64]. While our data did not specifically identify the source of IL-10, it is likely that antigen-presenting cells, particularly dendritic cells and monocytes, are the primary sources [65]. Previous studies have highlighted monocytes as the primary contributor of IL-10 in peripheral blood cells [66]. This shows that helminths are capable of modulating the immune system in such a way that the regulatory responses are enhanced [11,67,68,69,70,71,72]. The Th2-skewed immune response induced by helminth exposure has also been associated with tissue repair and remodeling, which could contribute to mitigating the damage to lung tissues caused by COVID-19 pneumonia [15,26,35,36]. Moreso, some helminths have been associated with the reduced risk or severity of certain comorbidities such as inflammatory bowel disease (IBD), autoimmune disorders, and allergic diseases [73]. If helminth exposure indirectly mitigates the severity of these conditions, individuals may have better overall health outcomes if they contract COVID-19 [74,75,76,77].

Our study’s strength lies in its capacity to address gaps in the current understanding of the interactions between helminth infections and COVID-19 clinical outcomes in African samples. It provides compelling evidence that exposure to helminth antigens confers protection against severe COVID-19. However, the main limitation of our work is that the samples were collected from a single African country. This geographic limitation may affect the generalizability of the results to other helminth-endemic areas or to the entire African continent. Additionally, while the correlations between helminth seropositivity and reduced severity of COVID-19 were observed in this cross-sectional cohort study, further investigations are needed to determine the causal relationships.

Although our findings are informative, our sample size of 104 participants is relatively small for drawing definitive conclusions. Future studies with larger sample sizes would provide more statistical power and reliable estimates of the observed effects thus enhancing the robustness and applicability of the findings to broader populations.

Our study used serological assays to detect helminth infections. Although serology is a useful tool for detecting past infections and exposure, it has limitations in discriminating between past and current infections especially in endemic countries [78]. Identification of patent helminth infection by serology is challenging in endemic areas because of long-lasting antibody responses and repeated exposure [79]. Despite these limitations, serological assays remain crucial in epidemiological studies and public health surveillance. Even though serology does not discriminate between past and current infections, it reveals the immune system’s reaction status to pathogens. For SARS-CoV-2 response mitigation, the presence of helminths, particularly those in the gut, is less important than the immune system’s reaction, which spills over to influence the response against SARS-CoV-2. This broadens the scope of our investigation, providing a valuable tool for analyzing the correlations between exposure to various helminths, current and past helminth infections, and COVID-19 severity.

The interaction between helminth infections and vaccine efficacy, particularly in the context of SARS-CoV-2, is a critical area of research that has implications for vaccination strategies in helminth-endemic regions. Our findings indicate a notable reduction in SARS-CoV-2-specific IgA and IgG levels, as well as diminished neutralizing antibody responses, in individuals exposed to helminths. This immune modulation is primarily due to a Th2-skewed immune response, characterized by elevated levels of cytokines such as IL-4 and IL-10, which are known to suppress Th1 responses and antibody production essential for effective vaccination against viral pathogens like SARS-CoV-2 [15,33,79]. The presence of helminth infections can significantly hinder the efficacy of vaccines that rely on robust Th1- and antibody-mediated immune responses. Studies have shown that chronic helminth infections can lead to decreased vaccine efficacy, as evidenced by the findings which indicate that established helminth infections at the time of vaccination adversely affect vaccine-specific immune responses [30,33]. This is particularly concerning in populations where helminth infections are prevalent, as the immune response necessary for effective vaccination may be compromised [79]. To mitigate these challenges, integrating adjunctive anti-helminthic treatments into vaccination campaigns could enhance vaccine responses. Evidence from previous studies suggests that deworming programs can improve vaccine efficacy, as demonstrated with BCG and influenza vaccines, where the removal of helminth infections resulted in enhanced immunogenicity and protective efficacy [33]. Furthermore, modified vaccine formulations that include adjuvants designed to enhance Th1 responses or counteract the effects of regulatory cytokines like IL-10 may also improve vaccine effectiveness in populations exposed to helminths [15,79]. Revised vaccination schedules could also play a role in ensuring adequate immunity in these populations. This might involve administering additional booster doses or utilizing higher antigen loads to overcome the immune suppression caused by helminth infections [33,79]. Our study underscores the necessity for vaccine trials that take into account co-endemic helminth infections, as this consideration is crucial for ensuring equitable vaccine efficacy in regions burdened by both infectious and parasitic diseases. Future research should aim to elucidate the specific mechanisms by which helminth exposure alters vaccine-induced immune responses, facilitating the development of tailored public health interventions that effectively address these challenges. In summary, the interplays between helminth infections and vaccine responses necessitate a multifaceted approach to vaccination strategies in endemic regions. By integrating anti-helminthic treatments, modifying vaccine formulations, and adjusting vaccination schedules, public health initiatives can enhance vaccine efficacy and improve health outcomes in populations affected by both viral and parasitic infections.

## 5. Conclusions

In conclusion, our study provides preliminary evidence that current or past helminth infections may modulate the immune response to SARS-CoV-2. This modulation appears to reduce Th1 and Th17 pro-inflammatory cytokine responses and the antibody response (IgA and IgG), resulting in diminished humoral immunity to SARS-CoV-2 and leading to better health outcomes. Our findings underscore the importance of considering helminth-induced immune modulation in disease control but also in vaccination strategies, as understanding these interactions could help us to optimize vaccine design and efficacy in populations with a high prevalence of helminth infections.

## Figures and Tables

**Figure 1 vaccines-13-00252-f001:**
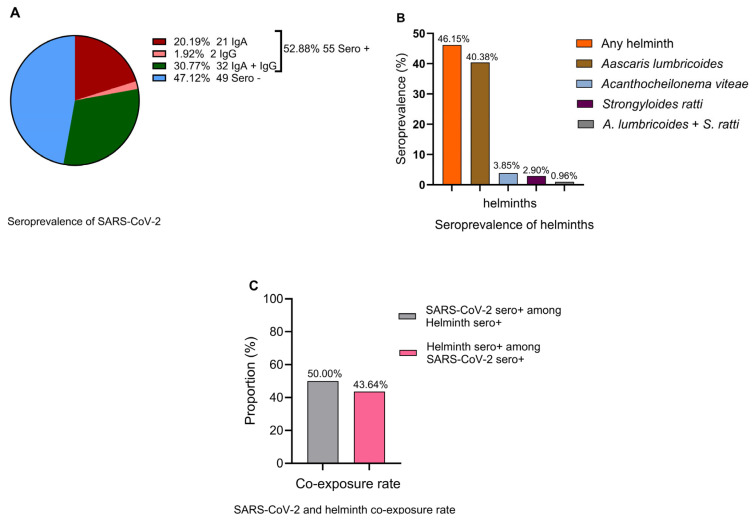
Seroprevalence of SARS-COV-2 (**A**) IgA (red), IgG (coral), IgA and IgG (green), and seronegative (blue), (**B**) any (pooled) helminth (orange), *Ascaris lumbricoides* (bronze), *Acanthocheilonema viteae* (homologous with human filaria species) (light blue), *Strongyloides ratti* (maroon), *A. lumbricoides* + *S. ratti* (gray), (**C**) SARS-CoV-2 seropositivity among helminth-seropositive individuals (gray) and helminth seropositivity among SARS-CoV-2-seropositive individuals (pink).

**Figure 2 vaccines-13-00252-f002:**
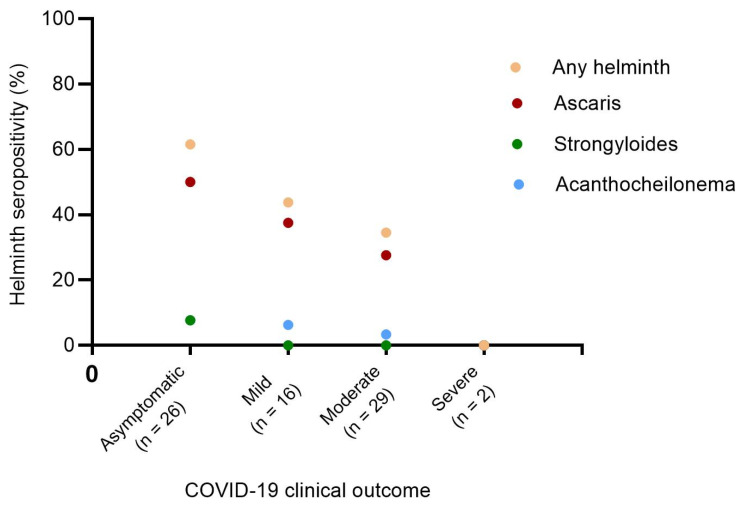
Prevalence of helminth seropositivity among COVID-19 patients with asymptomatic, mild, moderate, and severe clinical presentation. Chi-squared test *p*-values for trend *p* = 0.0197, *p* = 0.0465, *p* = 0.0890, and *p* = 0.4413 for any helminth (orange dots), *Ascaris lumbricoides* (red dots), *Acanthocheilonema viteae* (blue dots), and *Strongyloides ratti* (green dots) respectively. Any helminth refers to the pooled helminths (*Ascaris lumbricoides*, *Strongyloides ratti*, and *Acanthocheilonema viteae*).

**Figure 3 vaccines-13-00252-f003:**
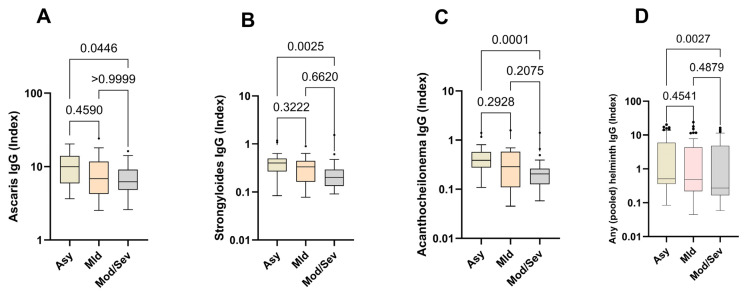
Higher expression of helminth-specific IgG in asymptomatic individuals. The graph represents the relative expression of IgG for *Ascaris lumbricoides* (**A**), *Strongyloides ratii* (**B**), *Acanthocheilonema viteae* (**C**), and pooled helminth (**D**) in the plasma of individuals who were either asymptomatic (Asy) (Wheat bars), mild (Mld) (pink bars) or moderate/severe (Mod/Sev) (gray bars) to SARS-CoV-2. The Kruskal–Wallis statistical test was utilized. Bars indicate the median with interquartile range, while the dots beyond the whiskers represent outliers.

**Figure 4 vaccines-13-00252-f004:**
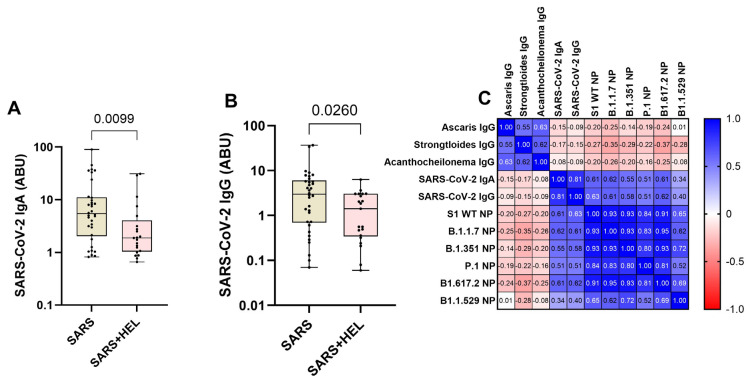
Lower expression of SARS-CoV-2-specific IgA/IgG and neutralizing potential (NP) in helminth-exposed individuals. Graphs (**A**,**B**) represent the relative expression of SARS-CoV-2-specific IgA (**A**) and IgG (**B**) in mono-exposed (wheat bars) and co-exposed individuals (pink bars). Bars indicate the median of immunoglobulin expression and interquartile range, while the dots represent individual participants. (**C**) also presents Spearman’s correlation between helminth-specific IgG and SARS-CoV-2 neutralizing antibodies against S1 WT, B.1.1.7, B1.351, B1.1.529, P.1, and B1.1.529 (**C**). Blue shades represent more positive correlations, while red shades represent more negative correlations, and white represents no correlation.

**Figure 5 vaccines-13-00252-f005:**
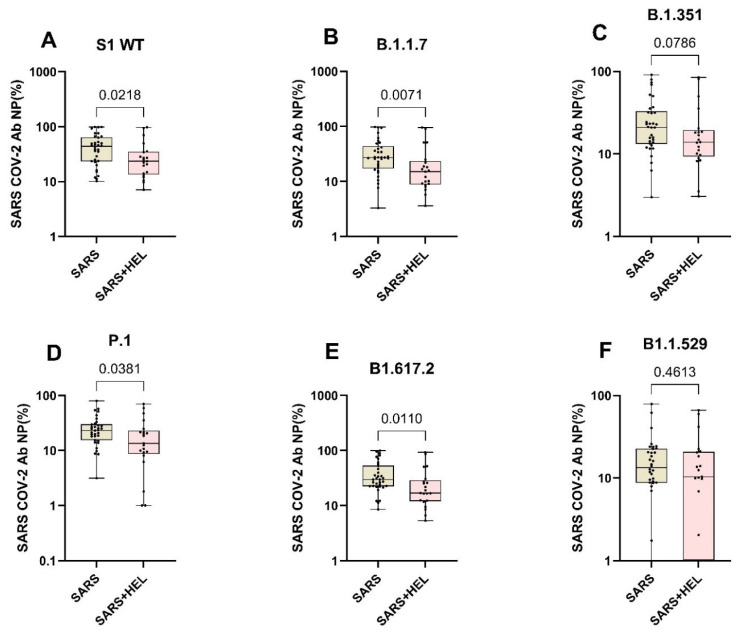
Expression of SARS-CoV-2 neutralizing antibodies against the wildtype and variants in mono- and co-exposed individuals. Graphs A to F represent the relative expression of SARS-CoV-2 neutralizing antibodies’ potential against the against S1 WT (**A**), B.1.1.7 (**B**), B1.351 (**C**), B1.1.529, P.1 (**D**), B1.617.2 (**E**) and B1.1.529 (**F**) in mono-exposed (wheat bars) and co-exposed individuals (pink bars). Analysis was generated using the Mann–Whitney U test. The dots represent individual participants. Bars represent median and interquartile range.

**Figure 6 vaccines-13-00252-f006:**
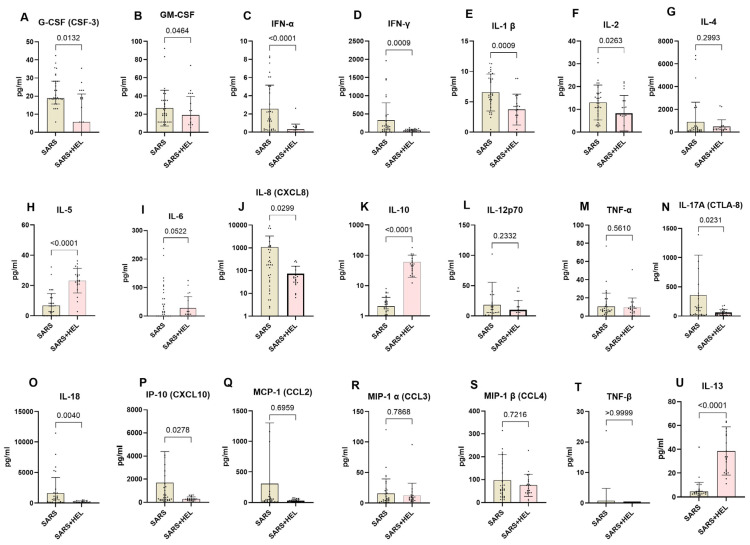
Lower expression of pro-inflammatory markers and higher expression of anti-inflammatory markers in co-exposed individuals. The graph represents the relative expression of G-CSF(**A**), GM-CSF (**B**), INF-α (**C**), IFN-γ (**D**), IL-1 β (**E**), IL-2 (**F**), IL-4 (**G**), IL-5 (**H**), IL-6 (**I**), IL-8 (**J**), IL-10 (**K**), IL-12p70 (**L**), TNF-α (**M**), IL-17A (**N**), IL-18 (**O**), IP-10 (**P**), MCP-1 (**Q**), MIP-1α (**R**), MIP-1β (**S**), TNF-β (**T**), IL-13 (**U**) in mono-exposed (wheat bars) and co-exposed individuals (pink bars). Bars indicate the median cytokine expression and interquartile range, while dots represent individual participants.

**Figure 7 vaccines-13-00252-f007:**
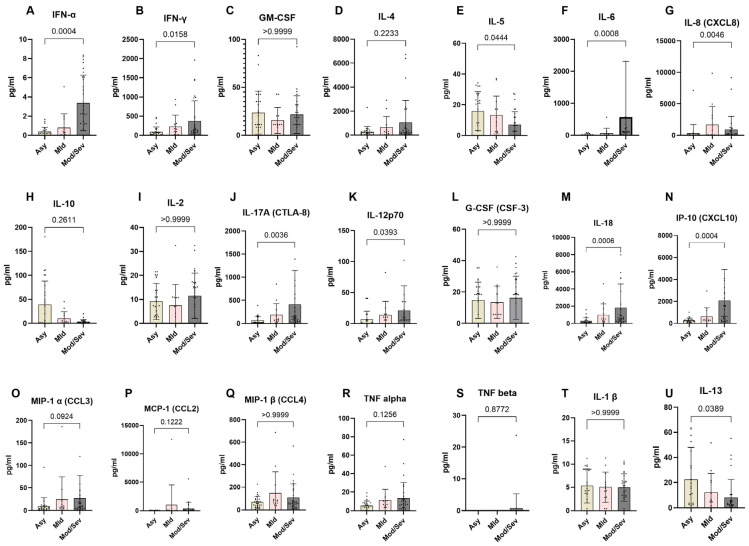
Lower expression of pro-inflammatory markers in asymptomatic individuals. The graph represents the relative expression of INF-α (**A**), IFN-γ(**B**), GM-CSF(**C**), IL-4 (**D**), IL-5 (**E**), IL-6 (**F**), IL-8 (**G**), IL-10 (**H**), IL-2 (**I**), IL-17A (**J**), IL-12p70 (**K**), G-CSF (**L**), IL-18 (**M**), IP-10 (**N**), MIP-1α (**O**), MCP-1 (**P**), MIP-1β (**Q**), TNF-α (**R**), TNF-β (**S**), IL-1β (**T**), and IL-13 (**U**) in individuals who were either asymptomatic (wheat bars), mild (pink bars), or moderate/severe (gray bars) to SARS-CoV-2 infection. The Kruskal–Wallis statistical test was utilized. Bars indicate the median cytokine expression and interquartile range, while the dots represent individual participants.

**Table 1 vaccines-13-00252-t001:** Demographic and clinical data of the study population.

	Number Examined n (%)	Clinical Outcome n (%)				
		Uninfected n = 31	Asymptomatic n = 26	Mild n = 16	Moderate n = 29	Severe n = 2
Gender						
Male	55 (52.9)	17 (54.8)	18 (69.2)	6 (37.5)	14 (48.3)	0 (0.0)
Female	49 (47.1)	14 (45.2)	8 (30.8)	10 (62.5)	15 (51.7)	2 (100.0)
Age group						
11–22	30 (28.8)	14 (45.2)	11 (42.2)	0 (0.0)	5 (17.2)	0 (0.0)
22–32	22 (21.2)	7 (22.6)	4 (15.4)	6 (37.5)	4 (13.8)	1 (50.0)
33–43	20 (19.2)	4 (12.9)	7 (26.9)	3 (18.8)	6 (20.7)	0 (0.0)
44–54	11 (10.6)	5 (16.1)	3 (11.5)	1 (6.5)	2 (6.9)	0 (0.0)
≥55	21 (20.2)	1 (3.2)	1 (3.8)	6 (37.5)	12 (41.4)	1 (50.0)

This table presents demographic and clinical information categorized by varying levels of COVID-19 severity: asymptomatic, mild, moderate and severe. Each category presents the total count of individuals, age classification, and gender; the sample size (n) of participants in the study is indicated along with percentages in brackets. The “number examined” column presents the relevant data for the entire sample.

**Table 2 vaccines-13-00252-t002:** Multivariate regression to explore the relationship between infectious status and COVID-19 clinical outcome.

Predictors	Estimate	aOR	Std. Error	Wald	df	*p*-Value	95% CI	
							L B	U B
Age	0.046	1.047	0.016	8.530	1	0.003	1.015	1.080
Gender (Female)	−0.946	0.388	0.571	2.751	1	0.097	0.127	1.188
*A. lumbricoides* seronegative	1.396	4.038	0.635	4.830	1	0.028	1.163	14.020
*A. viteae* seronegative	0.451	1.570	1.251	0.130	1	0.718	0.135	18.252
Helminth seronegative	1.415	4.115	0.596	5.628	1	0.018	1.279	13.242

This table presents the results of a multivariable regression analysis investigating factors associated with the severity of COVID-19. Variables include age, gender, *Ascaris lumbricoides* seroreactivity, *Acanthocheilonema viteae* seroreactivity, *Strongyloides ratti* seroreactivity, and any (pooled) helminth seroreactivity with their corresponding estimates, odds ratios (OR), standard errors, *p*-values, and 95% confidence intervals.

## Data Availability

Deidentified data not included in this manuscript can be obtained from the corresponding author upon reasonable request.

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
