# Peer review of "Helminth Seropositivity Inversely Correlated with Th1 and Th17 Cytokines and Severe COVID-19"

_vaccines, 2025, doi:10.3390/vaccines13030252_

Round 1

Reviewer 1 Report

Comments and Suggestions for Authors

The article by Brice A.N. Fogang and co-authors is devoted to such a topic as the influence of parasitic invasion on the severity of COVID-19. And the topic is really important and interesting, because many mechanisms of regulation of antiviral immunity, despite many years of research, remain unclear. And this also concerns the severity of a number of diseases caused by respiratory viruses.

The authors have done a great job. The set of methods is impressive, an analysis of the levels of IgA and IgG antibodies was carried out, as well as a quantitative analysis of the levels of a panel of cytokines and chemokines.

The authors correctly assessed the results obtained, indicating the limitations of the work.

I have several minor comments

1. There are errors in the article. So, a number of links are displayed incorrectly, for example, line 207, 227 and 250.

2. It is necessary to pay attention to Figure 1. The percentage signatures are very poorly visible in 1B of the figure.

Author Response

  1. There are errors in the article. So, a number of links are displayed incorrectly, for example, line 207, 227 and 250.

We thank the reviewer for pointing out the errors in the displayed links within the manuscript. Upon review, we identified that these issues were due to a software compatibility issue. To ensure consistency and avoid further formatting problems, we have deactivated all active links in the text and retained only the reference citations. Additionally, we have carefully corrected the issues and ensured that the corresponding references are correctly updated. Specifically, the revised version now includes the corrected links at lines 218, 236, 237, and 264.

  1. It is necessary to pay attention to Figure 1. The percentage signatures are very poorly visible in 1B of the figure.

We thank the reviewer for their observation regarding Figure 1B. We have adjusted the figure to enhance the visibility of the percentage labels, ensuring they are clearer and more readable in the revised version.

Reviewer 2 Report

Comments and Suggestions for Authors

A number of issues require further clarification or modification.

1.     The authors should include additional clinical data such as BMI, underlying diseases, and comorbidities in the supplementary table. More importantly, the authors should also specify the type of SARS-CoV-2 strain that infected each patient, as different strains can lead to varying clinical symptoms and neutralizing antibody levels.

2.     In Table 2, “no” is used to describe helminth serology reactions, which should be labeled as “negative” for clarity and consistency.

3.     Figure 3 presents three groups; however, a statistical analysis for the mild infection group is necessary to draw more comprehensive conclusions. Moreover, the small sample sizes of certain helminth groups make statistical analysis unreliable. Additionally, helminth-specific IgG levels are likely influenced by the timing of the infection, but the study does not provide adequate information about when the helminth infection occurred. 

4.     Similarly, Figure 4 compares the helminth seropositive and seronegative groups among COVID-19 patients, but there is no discussion of the timing of COVID-19 infection, which is critical when interpreting IgA/IgG levels.

5.     There are multiple errors in the references inserted in the article, and these should be corrected to ensure proper citation formatting.

Comments on the Quality of English Language

 Minor English revisions are required before proceeding with publication.

Author Response

  1. The authors should include additional clinical data such as BMI, underlying diseases, and comorbidities in the supplementary table. More importantly, the authors should also specify the type of SARS-CoV-2 strain that infected each patient, as different strains can lead to varying clinical symptoms and neutralizing antibody levels.

We thank the reviewer for their insightful comments. In our study, individuals with underlying conditions such as high blood pressure, malaria, HIV, and type 2 diabetes were excluded to minimize confounding factors that could influence immunological responses and disease outcomes. Pregnant women were also excluded due to the significant immunological changes associated with pregnancy (see lines 141–147). Although BMI data was not collected in our study, we have acknowledged its potential impact in the discussion (see lines 364–366).

Regarding the SARS-CoV-2 strain, our data was obtained during the third wave of the pandemic in Ghana, when the Delta variant (B.1.617.2) was the predominant strain in circulation (Morang’a et al., 2022). This information has now been specified in the revised manuscript (see updated lines 118–119).

References

Morang’a, C. M., Ngoi, J. M., Gyamfi, J., Amuzu, D. S. Y., Nuertey, B. D., Soglo, P. M., Appiah, V., Asante, I. A., Owusu-Oduro, P., Armoo, S., Adu-Gyasi, D., Amoako, N., Oliver-Commey, J., Owusu, M., Sylverken, A., Fenteng, E. D., M’cormack, V. V., Tei-Maya, F., Quansah, E. B., … Awandare, G. A. (2022). Genetic diversity of SARS-CoV-2 infections in Ghana from 2020-2021. Nature Communications, 13(1), 2494. https://doi.org/10.1038/S41467-022-30219-5

  1. In Table 2, “no” is used to describe helminth serology reactions, which should be labeled as “negative” for clarity and consistency.

We thank the reviewer for their observation regarding the labeling in Table 2. To improve clarity and consistency, we have replaced "no" with more precise terminology. Specifically, we have updated the descriptions to Ascaris lumbricoides seronegative, Acanthocheilonema viteae seronegative, and helminth seronegative. Please see the revised version of Table 2 for these changes.

  1. Figure 3 presents three groups; however, a statistical analysis for the mild infection group is necessary to draw more comprehensive conclusions. Moreover, the small sample sizes of certain helminth groups make statistical analysis unreliable. Additionally, helminth-specific IgG levels are likely influenced by the timing of the infection, but the study does not provide adequate information about when the helminth infection occurred. 

We thank the reviewer for their valuable feedback. As suggested, we have included the statistical analysis for the mild infection group. Please refer to Figure 3 in the revised manuscript for the updated analysis.

Regarding the small sample size, we have previously acknowledged this limitation in our manuscript. While our findings are informative, the sample size of 104 participants is relatively small for drawing definitive conclusions. We have emphasized that future studies with larger cohorts would provide greater statistical power and more reliable estimates of the observed effects (see lines 420–423).

Concerning the timing of helminth infections, we acknowledge that in endemic regions, continuous and repeated exposure makes it difficult to determine the precise timing of infection. However, antibody dynamics can provide insights into past exposure. We have clarified this in the discussion section (see lines 424–436).

We appreciate the reviewer’s insightful comments, which have helped improve the depth and clarity of our manuscript.

  1. Similarly, Figure 4 compares the helminth seropositive and seronegative groups among COVID-19 patients, but there is no discussion of the timing of COVID-19 infection, which is critical when interpreting IgA/IgG levels.

We thank the reviewer for this valuable observation. The timing of COVID-19 infection is indeed a critical aspect to consider. Individuals presenting with symptoms whether mild, moderate, or severe likely contracted the virus within the past 14 days (Lauer et al., 2020). However, determining the estimated timing of infection can be challenging, especially in asymptomatic individuals who may be unaware of their infection status and when they were exposed.

However, examining the expression of SARS-CoV-2-specific IgA and IgG can provide valuable insights into the stage of infection. Our findings indicate that the seroprevalence of SARS-CoV-2-specific IgA [20.19% (95% CI: 13.6% - 28.9%)] was significantly higher than that of SARS-CoV-2-specific IgG [1.9% (95% CI: 0.05% - 6.74%)] (Figure 1A). This pattern suggests recent or ongoing exposure to the virus, particularly at the mucosal surfaces where IgA is rapidly produced as an initial immune response. The relatively low IgG levels indicate that systemic immunity has not yet fully developed, which is commonly observed in the early stages of infection when the immune system prioritizes local containment of the pathogen. Kindly refer to lines 246–252 in the revised version, where we address how the stage of infection can influence antibody responses, specifically IgA and IgG, and their interpretation in the context of COVID-19 infection.

References

Lauer, S. A., Grantz, K. H., Bi, Q., Jones, F. K., Zheng, Q., Meredith, H. R., Azman, A. S., Reich, N. G., & Lessler, J. (2020). The Incubation Period of Coronavirus Disease 2019 (COVID-19) From Publicly Reported Confirmed Cases: Estimation and Application. Annals of Internal Medicine, 172(9), M20-0504. https://doi.org/10.7326/M20-0504

  1. There are multiple errors in the references inserted in the article, and these should be corrected to ensure proper citation formatting.

We thank the reviewer for the comment, The reference errors have been corrected in the article. Please refer to lines 218, 236, 237 and 264 in the revised manuscript, where the citations have been properly formatted and updated accordingly.

Comments on the Quality of English Language

 Minor English revisions are required before proceeding with publication.

We thank the reviewer for their feedback on the language quality. We have carefully revised the manuscript to improve clarity, grammar, and readability. Additionally, we have conducted a thorough proofreading to ensure linguistic accuracy. We appreciate the reviewer’s suggestion and have taken the necessary steps to enhance the overall quality of the manuscript.

Reviewer 3 Report

Comments and Suggestions for Authors

The author reports the finding of relationship of co-infection of helminth and SARS-CoV-2 and the expression level of cytokines that involved in the immune response process. The manuscript is well written with supporting results from experiment. However, there’re few points that need more explanation.

1)     The authors choose to analyze the co-infection of helminth and SARS-CoV-2 to make primary question for the research. However, will this cause the random connections? Why were helminth chosen?

2)     The question of prevalence of helminth and SARS-CoV-2 needs to be addressed for this cohort of study.

3)     Is there any other co-infection beside helminth and SARS-CoV-2 for those samples?

4)     The conclusion stating that “the infection of helminth from current or past” reduce the novelty of the manuscript. One can investigate the IgG, IgM expression level in other to deduce the stage of infection of particular virus. The author can investigate those value to have a better understanding of co-infection mechanisms.

Author Response

The author reports the finding of relationship of co-infection of helminth and SARS-CoV-2 and the expression level of cytokines that involved in the immune response process. The manuscript is well written with supporting results from experiment. However, there’re few points that need more explanation.

  • The authors choose to analyze the co-infection of helminth and SARS-CoV-2 to make primary question for the research. However, will this cause the random connections? Why were helminth chosen?

We thank the reviewer for their thoughtful question. The choice to analyze helminth and SARS-CoV-2 co-infection was based on prior evidence suggesting that chronic helminth infections can modulate host immune responses, potentially influencing susceptibility to and outcomes of other infections, including viral diseases. Specifically, helminths are known to modulate the immune system, which may alter disease severity or immune function during viral infections like COVID-19. The potential for helminths to impact inflammatory responses, cytokine profiles, and immune regulation made them a relevant choice for investigating their role in COVID-19 outcomes. Additionally, helminth infections are prevalent in many parts of the world, particularly in low- and middle-income countries, where COVID-19 outcomes could differ due to this co-infection. Thus, studying helminth co-infection in the context of global health provides valuable insights into how such infections may modify COVID-19 susceptibility and severity in these regions.

  • The question of prevalence of helminth and SARS-CoV-2 needs to be addressed for this cohort of study.

We thank the reviewer for their thoughtful question. In 2022, the seroprevalence of SARS-CoV-2 in Ghana was 67.10% (Donkor et al., 2023), while the prevalence was 3.6% (Asante et al., 2024). The World Health Organization (WHO) reported that the prevalence of soil-transmitted helminths in Ghana ranged from low (1–19.9%) to moderate (20–49.9%) (ESPEN,2024) The seroprevalence of helminths and SARS-CoV-2 in this cohort was 46.15% and 52.88% respectively. Please refer to Figure 1A, 1B and 1C, where the seroprevalence of SARS-CoV-2, helminths, and coinfections within the cohort is presented. This figure provides a clear overview of the distribution of SARS-CoV-2, helminths, and coinfections in the study participants.

References

Asante, I. A., Lwanga, C. N., Takyi, C., Sekyi-Yorke, A. N., Quarcoo, J. A., Odikro, M. A., Kploanyi, E. E., Donkor, I. O., Addo–Lartey, A., Duah, N. A., Odumang, D. A., Lomotey, E. S., Boatemaa, L., Kwasah, L., Nyarko, S. O., Affram, Y., Asiedu-Bekoe, F., & Kenu, E. (2024). Detection of SARS-CoV-2 Variants Imported Through Land Borders at the Height of the COVID-19 Pandemic in Ghana, 2022. Cureus, 16(8). https://doi.org/10.7759/CUREUS.68220

Donkor, I. O., Mensah, S. K., Dwomoh, D., Akorli, J., Abuaku, B., Ashong, Y., Opoku, M., Andoh, N. E., Sumboh, J. G., Ohene, S. A., Owusu-Asare, A. A., Quartey, J., Dumashie, E., Lomotey, E. S., Odumang, D. A., Gyamfi, G. O., Dorcoo, C., Afatodzie, M. S., Osabutey, D., … Koram, K. A. (2023). Modeling SARS-CoV-2 antibody seroprevalence and its determinants in Ghana: A nationally representative cross-sectional survey. PLOS Global Public Health, 3(5), e0001851. https://doi.org/10.1371/JOURNAL.PGPH.0001851

ESPEN (2024). https://espen.afro.who.int/system/files/content/maps/pdfs/MAP-Ghana-sth-iu-endemicity-2022_portrait.pdf. Accessed: 2025-01-23

  • Is there any other co-infection beside helminth and SARS-CoV-2 for those samples?

We thank the reviewer for their insightful comment. In our cohort, we excluded other potential coinfections and conditions such as type 2 diabetes, hypertension, pregnancy, malaria, and HIV which could potentially confound our study results, see lines 141-147. Our study focused on helminth co-infection with SARS-CoV-2, and our analysis was centred on understanding the interaction between these two infections. Therefore, other potential co-infections were excluded.

  • The conclusion stating that “the infection of helminth from current or past” reduce the novelty of the manuscript. One can investigate the IgG, IgM expression level in other to deduce the stage of infection of particular virus. The author can investigate those value to have a better understanding of co-infection mechanisms.

We thank the reviewer for their insightful comment. While IgM is typically a marker of recent or acute infections in viral and bacterial diseases, its utility in distinguishing between current and past helminth infections is limited for several reasons. Helminth infections are chronic and long-lasting, with continuous antigen exposure due to reinfection, leading to sustained IgG production rather than persistent IgM responses. In this context, IgG, particularly subclass patterns, is more informative for assessing long-term exposure and immune modulation. In addition, in helminth-endemic regions, like Ghana, individuals frequently encounter repeated infections throughout their lifetime. This can result in a mixed immune profile, where IgM responses may not necessarily indicate a new or ongoing infection but rather recent exposure, making interpretation difficult.

Round 2

Reviewer 2 Report

Comments and Suggestions for Authors

The authors have satisfactorily addressed all my comments and made the requested modifications to the manuscript, leading to significant improvement of the paper, which meets the high-quality standards of Vaccines.

Reviewer 3 Report

Comments and Suggestions for Authors

The authors addressed all the comments. The manuscript is now ready for publication.